# Maternal expression and breast milk transfer of an mRNA- encoded monoclonal antibody in a murine model of cholera

Jennifer E. Doering[1]☉, Yetunde Adewunmi[1]☉, Cailin E. Deal[2], Obadiah Plante[2], Andrea Carfi[2]\*, Nicholas J. Mantis🅞[1]\*

1 Division of Infectious Diseases, Wadsworth Center, New York State Department of Health, Albany, New York, United States of America, 2 Moderna, Inc., Cambridge, Massachusetts, United States of America

☉ These authors contributed equally to this work.
\* andrea.carfi@modernatx.com (AC); nicholas.mantis@health.ny.gov (NJM)

## Abstract

Breast milk confers infants with immunity to a multitude of pathogens reflective of prior maternal infections and vaccinations. However, in outbreak situations where infants may be vulnerable to lethal infections due to gaps in the maternal immune repertoire, a case can be made for supplementing breast milk with one or more pathogen-specific monoclonal antibodies (mAbs) with known prophylactic or therapeutic activity. As oral delivery of recombinant IgG and IgA mAbs to infants has proven challenging, we investigated the use of mRNA-lipid nanoparticle (LNP) technology to stimulate pathogen-specific mAbs in milk. mRNA encoding the *Vibrio cholerae* O1 specific mAb, ZAC-3, as a human IgG1 or dimeric IgA2, was encapsulated in lipid nanoparticles (LNP) and administered parenterally to lactating and non-lactating female mice. A single intravenous administration of mRNA-LNPs resulted in high and sustained expression of functional ZAC-3 IgG1 in the blood and breast milk of lactating dams. ZAC-3 IgA2 levels were lower and more transient. ZAC-3 IgG1 (but not IgA2) was also detected in the serum of suckling pups at levels proportional to those in the mothers, demonstrating successful transfer of functional antibodies to newborns. Levels of ZAC-3 IgG1 and IgA2 were not sufficient to limit intestinal colonization of *V. cholerae* O1 when pups were separated from dams following intragastric challenge; however, a significant reduction in bacterial burden was observed when challenged pups remained with dams for continuous breastfeeding. Our findings highlight the potential of mRNA-based mAb platforms in the maternal-newborn context, while acknowledging the need for optimized antibody isotypes, dosing, and tissue-specific delivery to improve mucosal immunity.

**Data availability statement:** All data are in the manuscript file.

**Funding:** This work was supported by funding from ModernaTx (Cambridge, MA, USA) to CED and OJP. CD, OP and AC are or were employees of ModernaTx at the time this work was conducted. The funders had no role in study design, data collection and analysis, decision to publish, or preparation of the manuscript.

**Competing interests:** I have read the journal's policy and the authors of this manuscript have the following competing interests: CED, AC, OJP are or were employees and shareholders in Moderna Inc. In addition, CED and OJP are co-inventors on international patent WO 2022/212191 A1. JED, YA, and NJM have declared that no competing interests exist.

## Author summary

For newborns, breastfeeding is one of the most effective defenses against diarrheal diseases. Human milk contains a myriad of bioactive compounds, including cytokines, defensins, lactoferrin, and milk oligosaccharides that collectively hinder opportunistic and pathogenic bacteria from colonizing the neonatal intestinal epithelium. Breast milk is also enriched in antibodies that afford immunity to newborns against enteric pathogens encountered in the first years of life. However, there are situations where supplementing breast milk with additional pathogen-specific antibodies could protect otherwise vulnerable newborns from newly emerged or rapidly evolving pathogens. In this report we investigate in a mouse model the potential of mRNA as a platform for delivery of human monoclonal antibodies into breast milk.

## Introduction

Worldwide, children under the age of five remain at risk of infectious diseases affecting the lower respiratory tract, bloodstream (neonatal sepsis), or gastrointestinal system (diarrhea) that can have long term consequences for health and development. Breast milk provides a crucial source of passive immunity during the early days, weeks and even months of life, as it contains an array of antibodies against viral and bacterial pathogens that the mother has encountered through infection or active vaccination [1]. For example, a recent examination of breast milk samples collected from women in five high and low-to-middle income countries revealed IgA and IgG antibodies against hundreds of pathogen-derived antigens [2]. The observed profiles are consistent with observations dating back decades that breastfed infants have reduced risk of infections, hospitalizations and morbidity as compared to non-breastfed cohorts [3]. Breast feeding also influences the gut microbiota and has health benefits on immune development [4]. Proactively modulating specific antibody titers in colostrum and breast milk in anticipation of potential pathogen exposures represents a powerful means to improve newborn health.

Augmenting pathogen-specific antibody levels in serum as well as colostrum and breast milk can be achieved through vaccination during pregnancy [5]. For instance, mothers who received influenza vaccines in the third trimester maintained high levels of IgA in their breast milk [6]. Likewise, abundant levels of pertussis-specific IgG and IgA are detected in breast milk after maternal vaccination [7,8]. In some cases, however, the levels of maternal antibodies passed to offspring are insufficient or decline too rapidly to afford protection [9]. Moreover, in the case of a disease outbreak or in instances where effective vaccines do not exist, alternative strategies for providing immunity to newborns is required. In such situations, monoclonal antibodies (mAbs) could provide immediate protective immunity.

There are pharmacokinetic and clinical pharmacology challenges associated with the use of mAbs directly in children such as dosing complexities, bio-distribution,

route of administration and immunogenicity [10,11]. Currently, only a few mAbs such as Palivizumab and Nirsevimab, which neutralize respiratory syncytial virus (RSV), are approved by the FDA for direct use in neonates [12]. This is because, generally, the use of mAbs in pediatrics is only explored after their safety and efficacy has been well established in adults. Given that maternal mAbs in breast milk can also attach to, and swiftly eliminate, pathogens found in breast-feeding infants, using breast milk as a vehicle to deliver pathogen-specific mAbs could provide a means to safely boost antibody levels by mimicking natural passive immunity [13].

The expression of mAbs from mRNA has demonstrated potential against diseases such as Chikungunya, HIV-1, and SARS CoV-2, with protective levels observed in mice, non-human primates and humans [14]. Therefore, in this study, we tested whether maternal administration of ZAC-3 mRNA-LNPs could induce functional antibody expression in serum and milk and enable transfer to offspring via breast milk which may support future strategies aimed at restricting pathogen replication. We employed cholera as a proof of concept, as it is a disease characterized by recurrent outbreaks that would benefit greatly from a passive immunotherapy strategy. ZAC-3 is a well-established protective mAb that targets the con-served core region of lipopolysaccharide (LPS) in *Vibrio cholerae* O1 strains. We evaluated the expression and transfer of mRNA encoding both ZAC-3 IgG1 and ZAC-3 IgA2 following systemic administration in lactating mice. While both isotypes were successfully expressed, IgA2 levels were substantially lower. Nonetheless, systemic delivery of mRNA-LNP resulted in robust expression of functional ZAC-3 IgG in maternal serum, with detectable transfer to both milk and the circulation of suckling pups.

## Methods

### Ethics statement

The mouse experiments described in this study were reviewed and approved by the Wadsworth Center's Institutional Animal Care and Use Committee (IACUC). The Wadsworth Center complies with the Public Health Service Policy on Humane Care and Use of Laboratory Animals and was issued assurance number A3183-01. The Wadsworth Center is fully accredited by the Association for Assessment and Accreditation of Laboratory Animal Care (AAALAC). Obtaining this voluntary accreditation status reflects that Wadsworth Center's Animal Care and Use Program meets all standards required by law and goes beyond the standards as it strives to achieve excellence in animal care and use. Per the National Institutes of Health's Office of Laboratory Animal Welfare (OLAW) guidelines, adult mice were euthanized by car-bon dioxide asphyxiation followed by cervical dislocation. Neonatal mice (<10 days of age) were subject to decapitation.

### Recombinant antibodies

Recombinant ZAC-3 IgG was produced by GeneArt (Thermo Fisher Scientific, Waltham, MA). ZAC-3 IgG was purified from transient transfection of a 2:1 ratio of HC to LC by weight in EXPI293 with HiTrap Protein A and polished using HiLoad Superdex 200 26/600 prep. ZAC-3 IgG concentrations were determined by absorption at 280nm using a Nano-Drop microvolume spectrophotometer (Thermo Fisher Scientific).

### Generation of modified mRNA and LNPs

Sequence-optimized mRNA encoding ZAC-3 IgG1 and IgA2 Fc regions were synthesized *in-vitro* using an optimized T7 RNA polymerase-mediated transcription reaction with complete replacement of uridine by N1-methyl-pseudouridine and a DNA template with open reading frames flanked by 5' untranslated region (UTR) and 3' UTR sequences with a terminal encoded polyA tail [15,16]. Lipid nanoparticle-formulated mRNA was produced through a modified ethanol-drop nanopre-cipitation process as described [17]. Briefly, ionizable, structural, helper and polyethylene glycol lipids were mixed with mRNA encoding immunoglobulin heavy chain (HC), light chain (LC), and where applicable the joining chain (JC) (2:1 HC to LC for IgG and 4:4:1 HC to LC to JC for IgA) in acetate buffer at a pH of 5.0 and at a ratio of 3:1 (lipids:mRNA). The

mixture was neutralized with Tris-Cl at a pH 7.5, sucrose was added as a cryoprotectant, and the final solution was sterile filtered. Vials were filled with formulated LNP and stored at -70°C. The drug product underwent analytical characterization, which included the determination of particle size and polydispersity, encapsulation, mRNA purity, osmolality, pH, endotoxin and bioburden, and deemed acceptable for *in-vivo* study.

## Bacterial strains and growth conditions

*V. cholerae* C6706 (El Tor Inaba) was routinely cultured in LB broth at 37°C with aeration. As necessary, media was supplemented with streptomycin (100 µg/ml) [18].

## Mouse protocols

Nonpregnant and timed pregnant female Swiss Webster (SW) mice aged 6–12 weeks were obtained from Taconic Biosciences (Germantown, NY). LNPs (1 mg/kg) or sterile PBS were administered by intravenous (IV) tail vein injection unless noted otherwise. Intravenous administration was used to achieve rapid and reproducible systemic expression of the encoded antibodies within the defined study window. A dose of 1 mg/kg mRNA-LNPs was selected solely based on prior studies demonstrating safety and tolerability of this dose in murine models, including no major adverse reactions, pathology, or mortality [16]. No physiological or behavioral abnormalities were observed in either dams or pups following mRNA-LNP administration, as assessed by routine daily monitoring of activity, nursing behavior, body condition, and survival.

ZAC-3 IgG1 and IgA2 expressions were assessed in serum collected from tail vein blood (n = 10 mice for experimental groups; n = 4 mice for control groups). For breast milk collection, dams (n = 3 mice per group) were separated from their pups for 2–4 h, anesthetized with isoflurane, then administered 2 IU of oxytocin subcutaneously. Milk was collected using a commercially available human breast pump (Spectra◊ S2 PlusÔ, Spectra Baby USA) modified with narrow-diameter tubing to enable connection to an 18-gauge needle hub inserted through a rubber stopper capping a 15 mL conical tube. A second needle, also fitted with tubing, was inserted through the same stopper and rotated between the teats of lactating mice to promote milk extraction. Milk was drawn into the conical tube via suction generated by the breast pump. Tubes were spun for 20 min at 1,500 x *g* at 4°C and the liquid portion (avoiding the fat layer on top and the cells on the bottom) was transferred to a fresh tube. A second centrifugation (13,200 x *g* for 15 min at 4°C) was performed before collecting the middle liquid portion. This final liquid was used for analysis. Total human IgG1 and IgA levels in serum and breast milk were determined using an isotype-specific ELISA, as detailed below.

## Capture enzyme-linked immunosorbent assays (ELISA) for human IgG and IgA

To quantitate human IgG and IgA in serum and breast milk samples, separate isotype-specific ELISAs were performed. Briefly, 384 well plates were coated with 0.02 mL per well of goat anti-human IgG or IgA Fc fragment (Bethyl Laboratories, Montgomery, TX) at 1:100 dilution in 0.05 M carbonate-bicarbonate buffer and incubated at room temperature for 3 h. The 384 well plates were then washed with 0.05% PBS-T and blocked overnight at 4°C with 0.05 mL per well of 2% goat serum in PBS-T. Serum and breast milk samples were serially diluted in 2% goat serum in PBS-T and transferred (0.02 mL per well) to the coated plates and incubated at room temperature for 90 min. Plates were then washed and incubated with goat anti-human IgG or IgA horseradish peroxidase secondary antibody (HRP; SouthernBiotech, Birmingham, AL; 1:5000) at room temperature for 1 h. Plates were subsequently washed and incubated with 0.02 mL per well of SureBlue TMB 1-C substrate (SeraCare, Gaithersburg, MD). The reaction was stopped with 0.02 mL per well of 1 M phosphoric acid and read at an absorbance of 450nm on a SpectraMax iD3 Microplate Reader (Molecular Devices, San Jose, CA). Absolute quantities of human antibody were extrapolated against a standard curve in GraphPad Prism 10.

## Bacterial whole cell ELISA

Immunolon 4HBX 96-well microtiter ELISA plates were coated overnight with 100 µL of *V. cholerae* C6706 at an $OD_{600}$ of 0.1. The plates were blocked for 2 h with 2% goat serum in PBS-T, washed and then treated with 100 µL serial dilutions of mouse serum samples. After a 1 h incubation at room temperature, the plates were washed and treated with 100 µL of goat anti-mouse HRP secondary antibody (1:2000 dilution; SouthernBiotech,) for 1 h at room temperature. ELISA plates were washed and developed using SureBlue Microwell Peroxidase Substrate. Plates were analyzed using a SpectraMax iD3 spectrophotometer with Softmax Pro 7.1 software (Molecular Devices).

## Complement-dependent serum bactericidal assay

The vibriocidal activity of IgG in sera and breast milk was determined using the agar plate vibriocidal assay [18]. Briefly, an overnight culture of *V. cholerae* C6706 was sub-cultured in LB then grown to mid-log at 37°C with aeration for 2 h. Mid-log phase cells were diluted to $4 \times 10^5$ CFUs/mL in PBS supplemented with guinea pig serum (20% vol/vol) then mixed 1:1 (vol/vol) in eppendorf tubes with pooled mouse milk (n = 3 mice/group) or serum samples (n = 4–10 mice/group) that had previously been incubated at 56°C for 30 min to inactivate endogenous complement. Tubes were then incubated at 37°C for 1 h and 50 µL aliquots were subsequently spread on LB agar plates after appropriate serial dilutions. The plates were incubated overnight at 37°C and CFUs were enumerated to determine the residual viability of *V. cholerae* C6706 that had been exposed to serum or milk samples in the presence of guinea pig complement. Percent vibriocidal activity was calculated by comparing the reduction in $\log_{10}$ CFUs in samples treated with either breast milk, lactating or non-lactating serum to those treated with naïve serum or breast milk, expressed as: [($\log_{10}$ CFU_naïve − $\log_{10}$ CFU_sample)/ $\log_{10}$ CFU_naïve] × 100.

## Neonatal intragastric challenge model and protection

To evaluate protective efficacy, pups (n = 5 per group) were gavaged per os with 0.05 mL of mid-log phase *V. cholerae* C6706 ($1 \times 10^8$ CFU/mL) two days (48 h) after their respective dams had received intravenous LNPs packaged with mRNA encoding either ZAC-3 IgG1 (n = 3 dams) or ZAC-3 IgA2 (n = 3 dams), or saline control (n = 2 dams). These experiments were performed three times. In the first run, pups were separated from dams for 24 h following gavage; in the subsequent two runs, pups were returned to their dams for continuous breastfeeding.

One day following *V. cholerae* challenge, the pups were euthanized by decapitation and blood collected immediately thereafter. The pups were dissected, and the small and large intestines were harvested and placed into 1 mL of sterile PBS, then homogenized (2 x 30 sec) using a Bead Mill 4 Homogenizer (Thermo Fisher Scientific). To enumerate bacterial CFUs in the homogenates, serial dilutions were plated on LB agar supplemented with streptomycin (100 µg/mL) then incubated overnight at 37°C. Intestinal bacterial counts recovered 24 h after challenge were used as the measure of protection. Blood and breast milk samples were collected from dams on the same day that pups were challenged with *V. cholerae*.

## Statistical analysis

Statistical analysis and graphics were performed using GraphPad Prism version 10.4 (Boston, MA). Differences between groups were determined using an ordinary two-way ANOVA with Sidak's multiple comparison test (P < 0.05), after verifying normality of data distribution using the Shapiro–Wilk test. Statistical testing was performed only for comparisons involving independent biological replicates (e.g., lactating vs. non-lactating mice in Fig 2B). Assays that used pooled samples or represented descriptive presence/absence measurements were not subjected to hypothesis testing, as pooled data do not represent independent observations. For data sets involving two groups, significance was assessed by performing an unpaired t test without Welch's correction after verifying that variances between groups were equal.

## Results

### Detection of functional mRNA-LNP encoded ZAC-3 in the circulation of non-pregnant and pregnant mice

To determine if ZAC-3 IgG1 and IgA2 are expressed from an mRNA template *in vivo,* we administered nucleoside-modified ZAC-3 IgG1 and IgA2 mRNA LNPs (1 mg/kg) to adult female SW mice intravenously and monitored serum samples for the presence of total human and *V. cholerae*-specific IgG and IgA over the course of one week (Fig 1A). Human IgG was detected in all mice at all time points examined, with peak concentrations (>200 µg/mL) at 48 h (Fig 1B). IgA2 was also present in serum, though its concentration was at least 15-fold lower than that of IgG between 24 h and 72 h and could not be detected at 168 h (Fig 1D). Neither IgG1 nor IgA2 was detected in sham treated animals (Fig 1B and 1D). A bacterial whole-cell ELISA confirmed that human IgG1 in serum bound to immobilized *V. cholerae* C6706 (Ogawa) (Fig 1C), while for IgA binding was detected at 24 h but declined at later time points (Fig 1E). Moreover, mice containing *V. cholerae*-specific IgG each exhibited complement-dependent vibriocidal activity when tested at a dilution corresponding to 2 µg/mL, while sera from naïve mice had no detectable activity (Fig 1F). The mRNA-encoded ZAC-3 IgG was comparable in potency to recombinant ZAC-3 IgG (Fig 1F). Serum samples containing ZAC-3 IgA had no detectable complement-dependent vibriocidal activity, as expected.

### ZAC-3 distribution in lactating and non-lactating mice

In anticipation of studies examining maternal transfer of ZAC-3, we next compared ZAC-3 IgG1 and IgA levels in the sera of lactating mice and non-lactating mice at four different timepoints (24, 48, 72 and 168 h) (Fig 2A). At the time points examined, ZAC-3 IgG1 was five times higher in the sera of non-lactating mice as compared to lactating mice (Fig 2B). For ZAC-3 IgA2, levels were at least 2.5 times higher in non-lactating mice at all time points with a peak of 4.5-fold difference at the 48-h time point between lactating and non-lactating mice (Fig 2C). The observed differences in IgG antibody concentrations were also reflected in the vibriocidal activity of the serum samples, as undiluted sera from non-lactating mice were on average 40% more efficient at killing *V. cholerae* C6706 than sera from lactating dams (Fig 2D). We speculate that, in lactating mice, a portion of ZAC-3 IgG1 and IgA2 may be shunted from circulation to breast milk, thereby accounting for the relative differences in antibody levels.

### Detection of functional mRNA-LNP encoded ZAC-3 in mouse milk

To examine the distribution of ZAC-3 in milk, milk was collected from dams at 24 h through 96 h after mRNA-LNP administration. The whey fraction was evaluated for functional ZAC-3 IgG and IgA2 (Fig 3A). ZAC-3 IgG1 levels increased progressively with time, peaking at 2 µg/mL at 96 h post-administration. ZAC-3 IgG levels in milk were ~five times less than in serum (Figs 2B and 3B). Milk-derived IgG antibody bound to immobilized *V. cholerae* C6706 at levels proportional to milk concentrations (Fig 3C). Moreover, we detected vibriocidal activity in pooled milk collected from dams 96 h after ZAC-3 mRNA-LNP treatment when the ZAC-3 IgG antibody concentration was the highest. The vibriocidal activity against *V. cholerae* C6706 in experimental milk samples was 100-fold higher than naïve milk samples (Fig 3F). ZAC-3 IgA was also detected in breast milk, peaking at 48 h with an average of ~1 µg/mL (Fig 3D). Accordingly, IgA levels in milk were ~10-times lower than in serum (Figs 2D and 3D). IgA2 binding to immobilized *V. cholerae* C6706 also occurred at 24 h and 48 h but not at 72 h or 96 h consistent with the markedly reduced IgA concentrations at these later time points. (Fig 3E). Nonethless, the fact that ZAC-3 IgG and IgA were detectable in breast mile warranted investigation as to whether antibodies could be transferred to pups.

### Maternal transfer of functional ZAC-3 IgG1 to suckling mice

Next, we investigated the extent of antibody transfer between dams and pups over the course of 7 days by analyzing serum IgG and IgA2 collected from pups that fed from vaccinated dams (Fig 4A). It is known that milk-derived antibodies

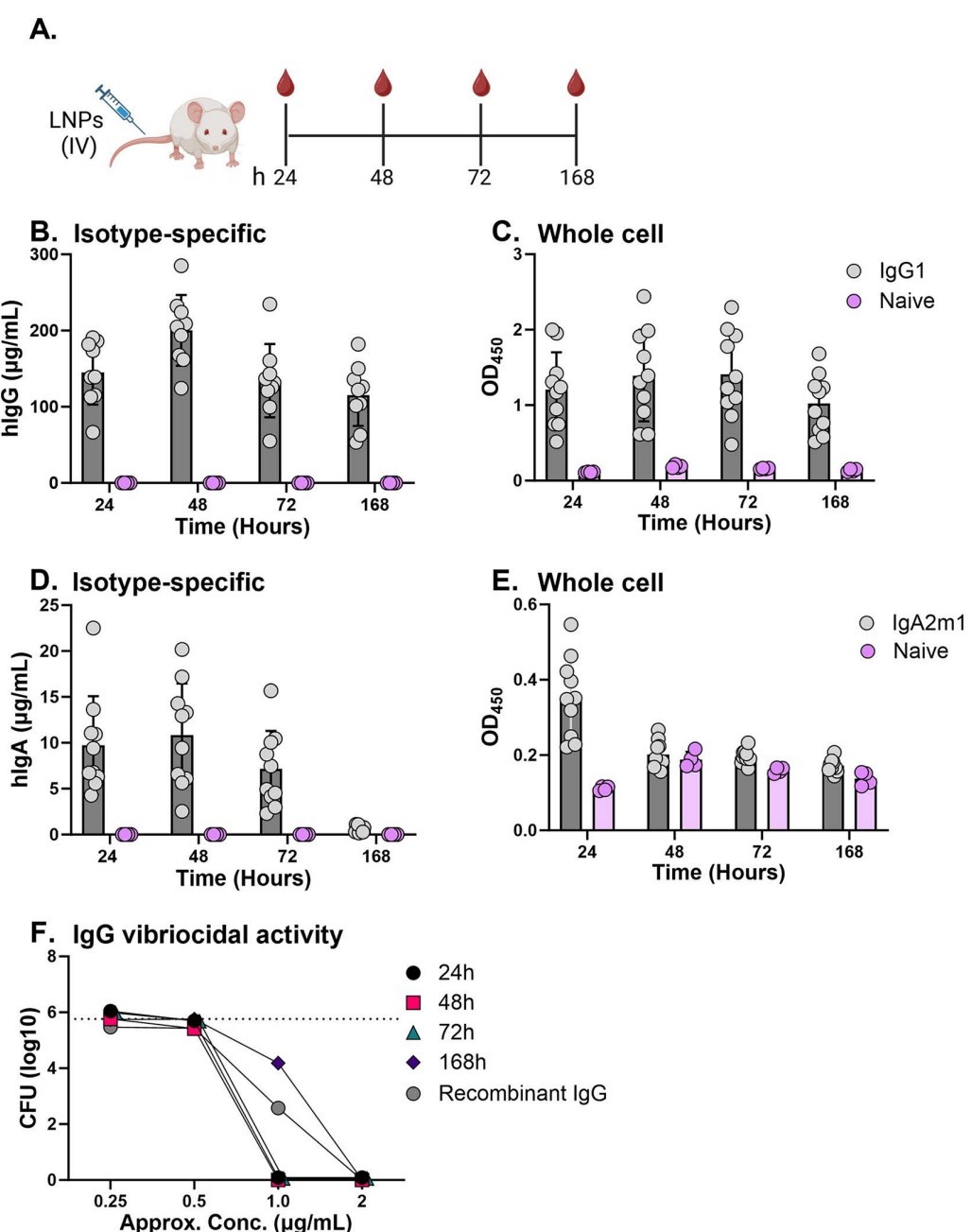

**Fig 1. Functional mRNA-encoded ZAC-3 IgG1 and IgA2 in serum. (A)** Cartoon depiction of experimental timeline. Adult female SW mice were intravenously administered ZAC-3 encoding mRNA-LNP (1 mg/kg) or PBS. Serum samples were collected 24, 48, 72, and 168 h later. **(B)** Total hIgG (µg/mL) in serum at the indicated time points, as determined by an isotype-specific ELISA. **(C)** *V. cholerae* C6706 specific IgG binding was assessed by whole-cell ELISA (serum used at a 1:50 dilution corresponding to hIgG concentrations of 3, 4, 3 and 2.5 µg/mL at 24, 48, 72 and 168 h, respectively. **(D)** Total hIgA (µg/mL) in serum at the indicated time points, as determined by an isotype-specific ELISA. **(E)** *V. cholerae* C6706 specific IgA binding was assessed by whole-cell ELISA (serum used at a 1:50 dilution corresponding to hIgA concentrations of 0.2, 0.2, 0.16 and 0 µg/mL at 24, 48, 72 and 168 h, respectively. For panels **B**, **C**, **D**, and **E** each symbol (gray or pink circles) represents a single mouse. The bars represent the mean±SD per group (n=10 mice for experimental groups; n=4 mice for control groups). **(F)** Complement-dependent vibriocidal activity in corresponding pooled serum samples from indicated time points, as described in the Methods section. The dashed line represents the average CFUs recovered from control sera at equivalent dilutions to experimental samples. Panel A created in BioRender: Adewunmi, Y. (2026) https://BioRender.com/ifii944.

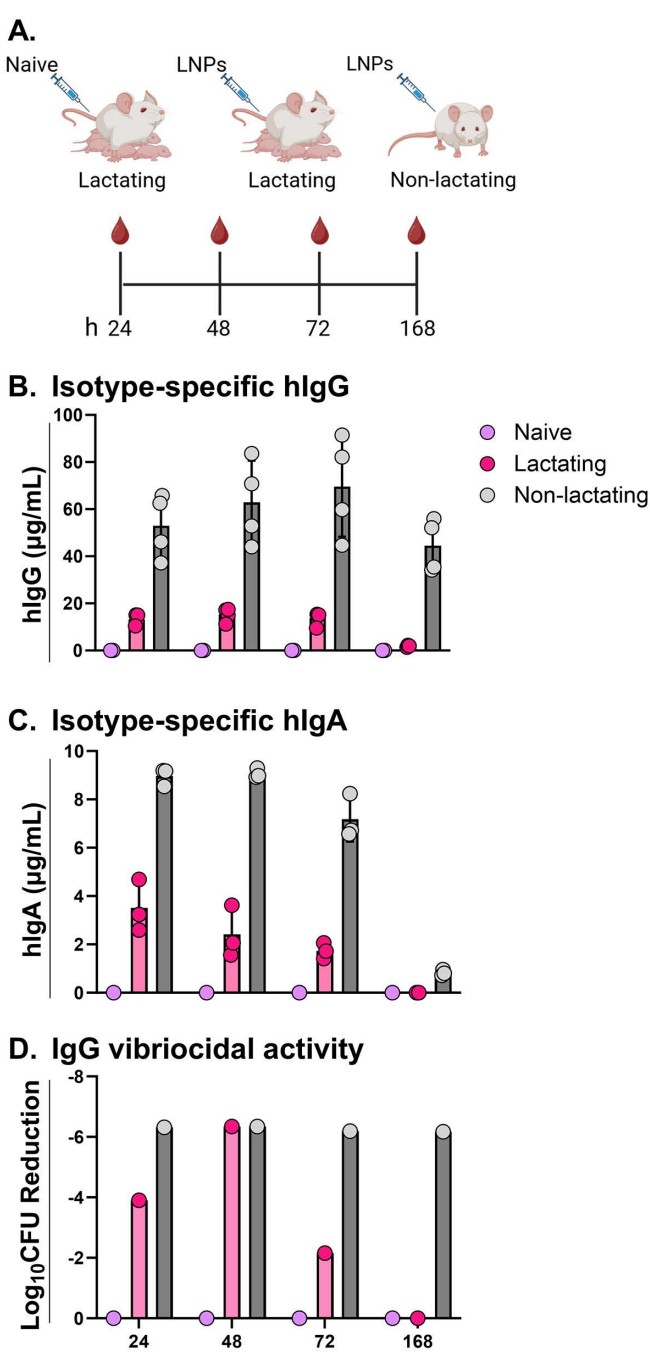

**Fig 2. Mouse lactation status influences mRNA-LNP encoded ZAC-3 IgG and IgA serum levels.** Lactating and non-lactating SW mice were intra-venously administered LNPs (1 mg/kg) or PBS on postpartum days 3-7. **(A)** Cartoon depicting experimental design in which serum was collected at 24, 48, 72, and 168 h. **(B)** Serum hIgG levels in naïve (lavender circles), lactating (red circles) and non-lactating (gray circles) mice, as determined using an isotype-specific ELISA, at indicated time points. Each symbol indicates a single mouse. **(C)** Serum hIgA levels in naïve (lavender circles), lactating (red circles) and non-lactating (gray circles) mice, as determined using an isotype-specific ELISA, at indicated time points. Each symbol indicates a single mouse. **(D)** Complement-dependent, vibriocidal activity of serum samples within naïve (lavender circles), lactating (red circles) and non-lactating (gray circles) mice. *V. cholerae* C6706, expressed as $Log_{10}$ CFU reduction. Pooled serum samples used in the vibriocidal assay were taken at the same time points and at concentrations corresponding to the average hIgG levels shown in panel B. Each symbol represents one pooled sample per time point. Bars represent mean ± SD per group (n = 4 mice per group). Panel A created in BioRender: Adewunmi, Y. (2026) https://BioRender.com/2aphrkh.

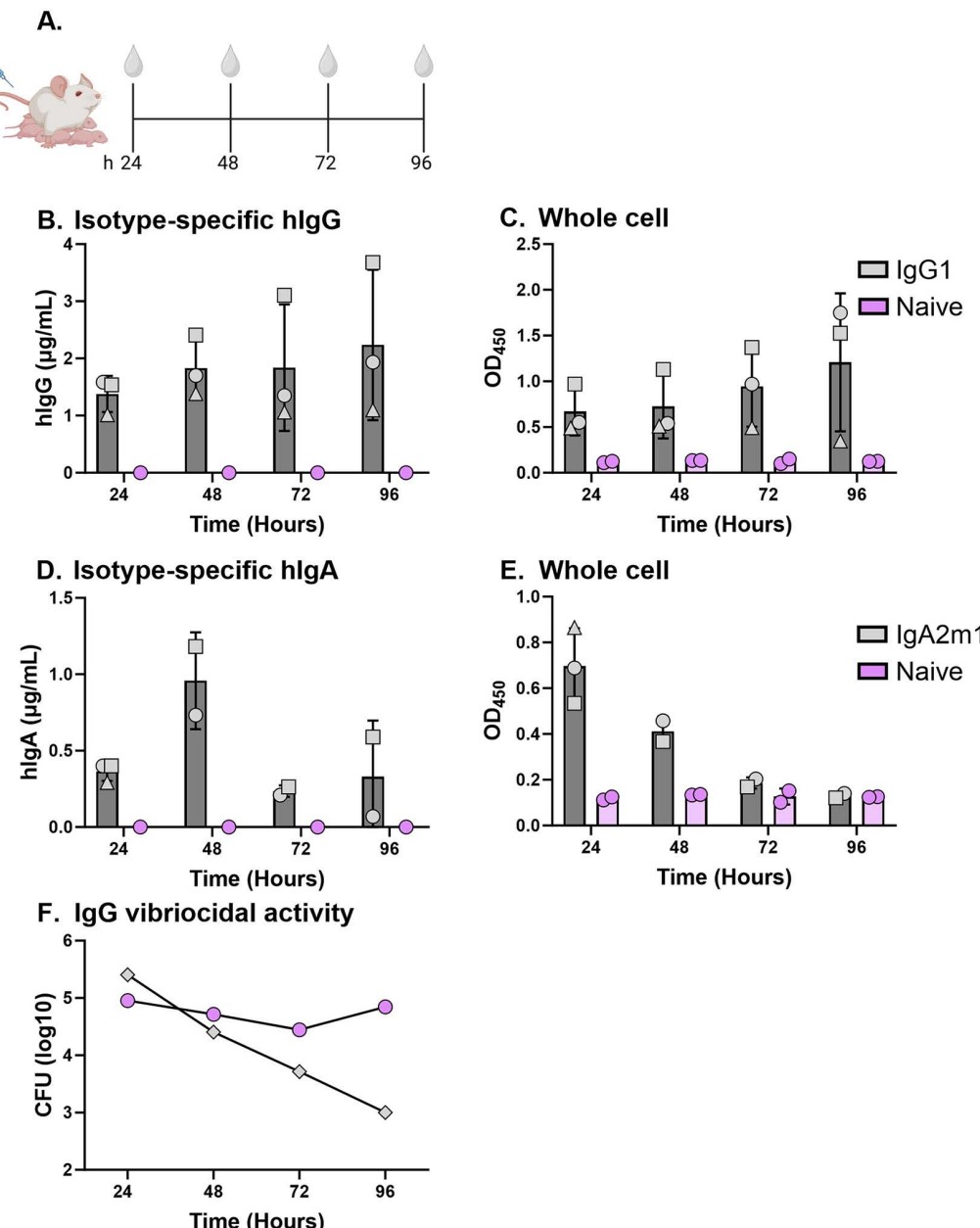

**Fig 3. Functional ZAC-3 IgG1 and IgA in breast milk.** SW mice were intravenously administered 1 mg/kg of LNPs or PBS 3-7 days after giving birth. **(A)** Cartoon depicting experimental design in which milk was collected at 24, 48, 72, and 168 h. **(B)** Concentration of hIgG detected in mice breast milk, as determined by an isotype-specific ELISA. Milk was collected at the indicated time points from three individual lactating dams (indicated by circle, square and triangle). **(C)** *V. cholerae*-specific IgG in milk was assessed by whole-cell ELISA. Milk collected at 24, 48, 72, and 96 h was tested at a fixed 1:4 dilution, which correspond to 0.2, 0.2, 0.25, and 0.3 μg/mL of ZAC-3 IgG, respectively. **(D)** Concentration of hIgA2 detected in mice breast milk, as determined by an isotype-specific ELISA. Milk was collected at the indicated time points from three individual lactating dams (indicated by circle, square and triangle). **(E)** *V. cholerae*-specific IgA2 in milk was assessed by whole-cell ELISA. Milk collected at 24, 48, 72, and 96 h was tested at a fixed 1:4 dilution, which correspond to 0.1, 0.4, 0.08, and 0.1 μg/mL of ZAC-3 IgA2, respectively. **(F)** Vibriocidal activity of pooled milk samples from each dam. The concentration of ZAC-3 IgG in the pooled samples corresponding to the average hIgG levels shown in panel B. Each group represents the mean±SD of 2-3 mice per group. Panel A created in BioRender: Adewunmi, Y. (2026) https://BioRender.com/ea34xua.

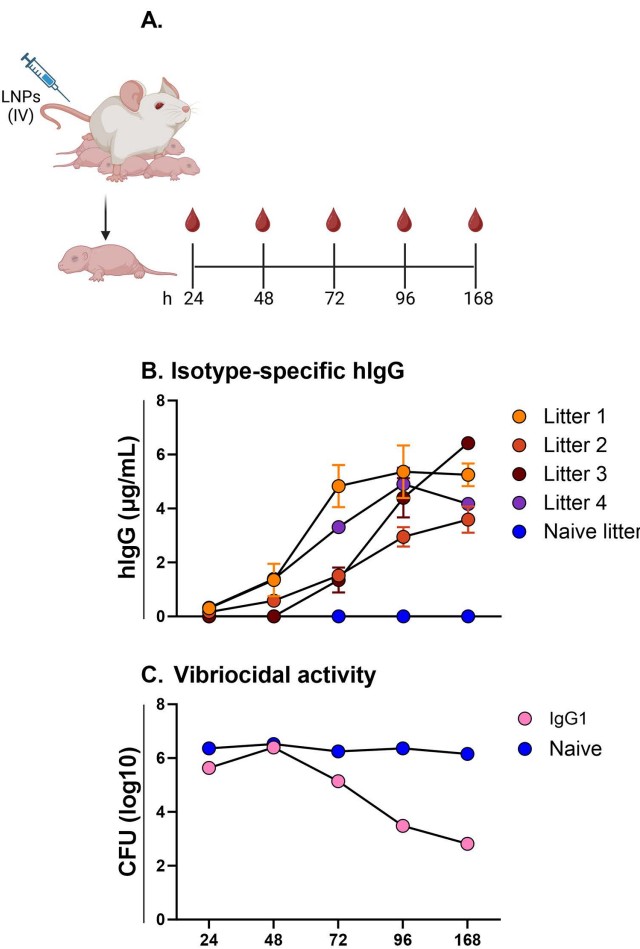

**A.**

LNPs (IV)

h 24   48   72   96   168

**B. Isotype-specific hIgG**

hIgG (µg/mL)

- Litter 1
- Litter 2
- Litter 3
- Litter 4
- Naive litter

**C. Vibriocidal activity**

CFU (log10)

- IgG1
- Naive

24   48   72   96   168

**Time (hours)**

**Fig 4. Maternally-derived ZAC-3 IgG accumulates in the serum of suckling mice.** ZAC-3 mRNA-LNPs (1 mg/kg) or PBS was administered intravenously to dams 3-7 days after giving birth. **(A)** Cartoon depicting experimental design in which serum was collected from pups at 24, 48, 72, and 168 h from four different experimental litters and two naïve litters. The day of mRNA-LNP administration was considered time 0. **(B)** Human IgG in serum, as determined using an isotype-specific capture ELISA. For each time point, a symbol represents values from two pups. **(C)** Vibriocidal activity of pup sera over the course of 168 h. For each time point, sera from all litters were pooled and used in the vibriocidal assay with average concentrations corresponding to hIgG concentrations of 1.5, 2, 2, and 2.5 µg/mL at 24, 48, 72, and 96 h, as described previously. Pools were from 2 mice per litter with two naïve litters and four ZAC-3 mRNA-LNP litters. Panel A created in BioRender: Adewunmi, Y. (2026) https://BioRender.com/8g0r3s2.

can accumulate in the serum of suckling rodents [1]. In agreement with this, we observed that serum ZAC-3 IgG1 levels in the pups increased with time, plateauing at ~4.5 µg/mL at 168 h (Fig 4B). While ZAC-3 IgG concentrations varied from litter to litter (Fig 4B), the amounts in the sera of pups were generally reflective of levels in the milk of their respective dams. Consistent with the low IgA2 concentrations detected in breast milk (Fig 3), pups that fed from dams that were vaccinated with ZAC-3 IgA2 encoding mRNA, had no detectable levels of ZAC-3 IgA2 in their serum. Vibriocidal activity against *V. cholerae* C6706 was detected in pooled IgG sera from each litter at timepoints between 24 and 168 h (Fig 4C). This result confirms that ZAC-3 IgG retains functionality following transfer from dam to pup through ingested breast milk and vibriocidal activity is dependent upon concentration.

## Functional activity of ZAC-3 IgG and IgA *in vivo*

To assess whether ZAC-3 IgG and IgA achieved levels sufficient to impact bacterial colonization in pups, suckling mice from control or ZAC-3 mRNA-LNP treated dams were intragastrically challenged with 5 x 10$^6$ CFUs of *V. cholerae* C6706 (Fig 5A). The neonatal mice were then separated from their dams and sacrificed 24 h later. Our results confirmed that ZAC-3 IgG and IgA were expressed in the serum and milk of all dams that received both ZAC-3 IgG and IgA mRNA-LNPs (Fig 5B and 5D); with substantially lower levels observed in IgA-treated dams (Fig 5D). The amount of ZAC-3 IgG in milk was approximately only 20% of that found in serum (Fig 5B) and ranged between 13–25% for IgA (Fig 5D). ZAC-3 IgG was also detected in the sera of suckling mice (Fig 5B) but not in the sera of pups that suckled from ZAC-3 IgA injected dams (Fig 5D).

Despite the presence of ZAC-3 IgG in the sera of the pups, the number of *V. cholerae* C6706 CFUs recovered from intestinal homogenates of neonatal animals following intragastric challenge was similar between control and mRNA LNP-treated groups of mice (Fig 5C). We observed similar findings in IgA breast-fed pups (Fig 5E); suggesting that ZAC-3 IgG and IgA levels in the gut did not reach the threshold required for protection.

Notably, ZAC-3 IgG and IgA were not detected in gut tissue homogenates derived from suckling pups, possibly because the antibodies were rapidly absorbed into circulation or were below the limit of detection for our binding assay in pups that were separated from dams. Thus, the *in vivo* functional activity of ZAC-3 in this specific model could not be ascertained.

Since the *in vivo* functionality of ZAC-3 in the model involving separation of pups from dams could not be determined, we repeated this study but returned pups to their dams after oral gavage with *V. cholerae* to assess whether continuous breastfeeding following infection would reveal ZAC-3 functional activity, as ongoing milk intake could increase the likelihood that ZAC-3 IgG and IgA encounter *V. cholerae* in the intestinal lumen before transport from the digestive tract into circulation (Fig 6A). Indeed, across two independent experiments, we observed a significant reduction (~1.5 fold) in the levels of *V. cholerae* C6706 in pups that fed from ZAC-3 IgG vaccinated dams compared to naïve-treated animals (Fig 6B). Similarly, a reduction in the levels of *V. cholerae* C6706 was also observed in pups that fed from ZAC-3 IgA vaccinated dams relative to naïve-treated animals, albeit to a lesser degree compared to IgG (Fig 6C), which is consistent with the overall lower levels of ZAC-IgA compared to IgG throughout the study.

## Discussion

The protection afforded by passive immunity has saved countless numbers of infants across the globe. For instance, SARS-CoV-2 mRNA vaccinations during and after pregnancy conferred significant protection to newborns and reduced hospitalizations through transplacental antibody transfer and/or breast milk derived neutralizing antibodies [19]. Several studies have shown that at least 4 weeks are required to derive sufficient levels of SARS-CoV-2 neutralizing antibodies in breast milk because an adaptive immune response had to be generated in the mother following vaccination [20]. However, this may not be rapid enough to protect newborns against infections in emergency situations. The use of mRNA-mediated antibody therapy represents a potential strategy for the delivery of robust mAb titers to newborns via breast milk as it eliminates the need for the generation of an adaptive immune response in the mother and overcomes the safety challenges associated with the direct administration of mAbs in the infant [21].

By employing a previously established mouse model and the well-studied anti-core LPS *V. cholerae* antibody [22], our study demonstrates that mRNA-encoded IgG1 mAb can be expressed in a mother and transferred to her suckling pup via breast milk and potentially confer protection against cholera following continuous breastfeeding. We confirmed that systemic delivery of mRNA-LNPs enabled the *in vivo* synthesis and expression of the encoded ZAC-3 IgA and IgG antibody in the serum of female mice as early as one day after mRNA-LNP treatment, with levels sustained for up to 3 and 7 days, respectively. Additionally, ZAC-3 IgG and IgA was detected in breast milk collected 24 h after mRNA-LNP administration,

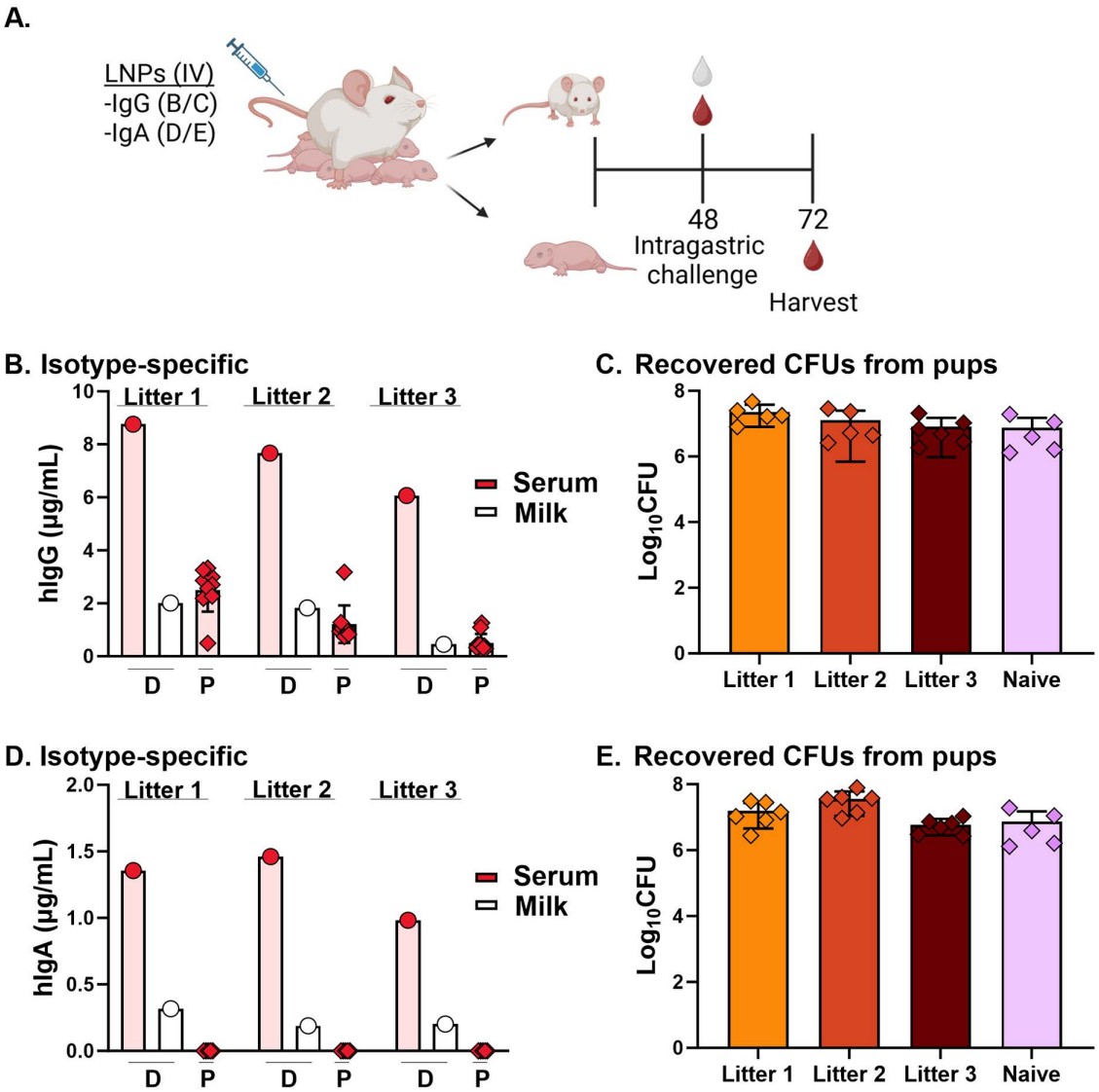

**Fig 5. Capacity of ZAC-3 IgG1 and IgA to reduce *V. cholerae* colonization. (A)** Lactating SW mice were intravenously administered with LNPs (1 mg/kg; N = 3) or PBS (N = 2) 3-7 days after giving birth. This was designated as study time 0. 48 h post-administration, breast-fed pups were intragastrically challenged with *V. cholerae* (1 x 10⁵ CFUs). Serum and milk were collected from ZAC-3 treated (n = 3) or naïve dams (n = 2) at 48 h, while serum from all pups (n = 10/group) was collected at 72 h. Pups were euthanized and small and large intestines were harvested, homogenized, and plated for CFUs (n = 5). **(B)** Data represents the concentration of hIgG in dam serum (pink bars with red circle symbols), milk (white bars with white circle symbols) and pup serum (pink bars with red diamond symbols), assessed using isotype-specific ELISA. Dam serum and milk samples are denoted by "D" under the appropriate bars and pup serum samples are denoted by "P". **(C)** Data represents recovered CFUs of *V. cholerae* in the intestinal homogenates from each litter in the ZAC-3 IgG treatment and naïve groups. **(D)** Data represents the concentration of hIgA in dam serum (pink bars with red circle symbols), milk (white bars with white circle symbols) and pup serum (pink bars with red diamond symbols), assessed using isotype-specific ELISA. Dam serum and milk samples are denoted by "D" under the appropriate bars and pup serum samples are denoted by "P". **(E)** Data represents recovered CFUs of *V. cholerae* in the intestinal homogenates from each litter in the ZAC-3 IgA treatment and naïve groups. Bars for dams represent an individual animal, while bars for pups show the mean ± SD of the group. Panel A was created in BioRender. Adewunmi, Y. (2026) https://BioRender.com/s16efvl.

**A.**

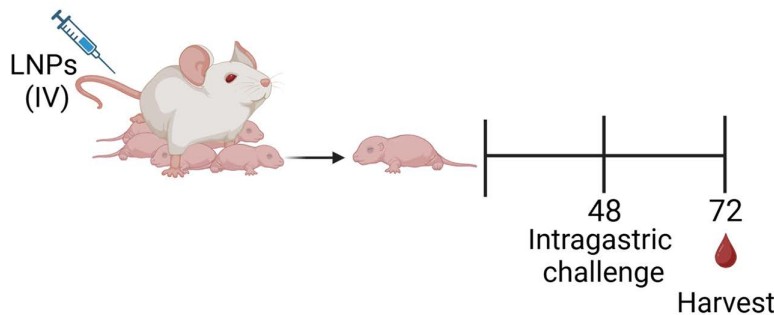

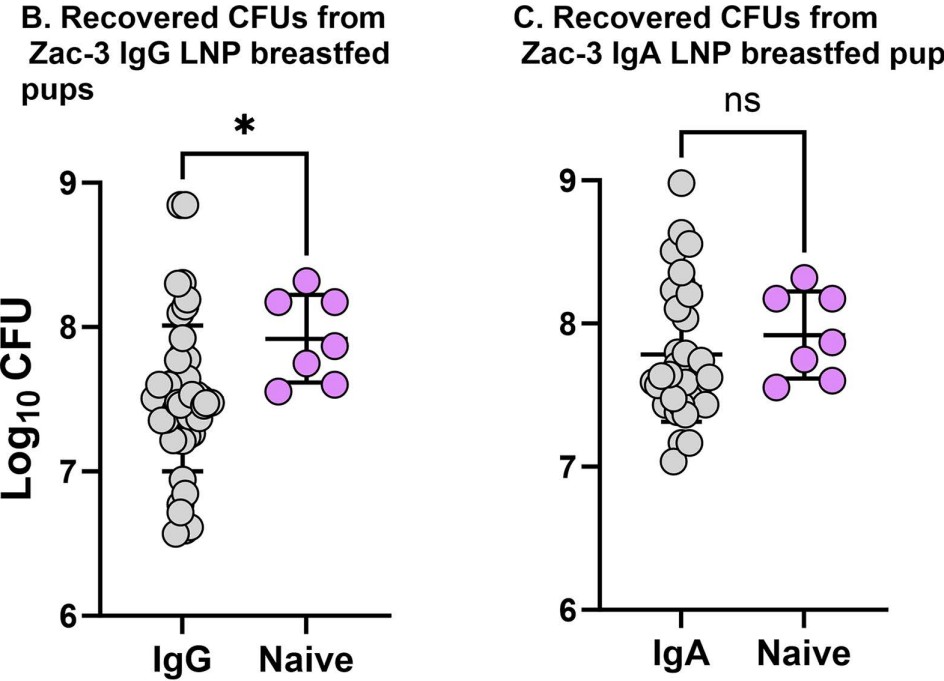

**B. Recovered CFUs from Zac-3 IgG LNP breastfed pups**

**C. Recovered CFUs from Zac-3 IgA LNP breastfed pups**

**Fig 6. Continuous breastfeeding improves *V. cholerae* reduction in pup guts. (A)** Lactating SW mice were intravenously administered with LNPs (1 mg/kg; N = 3) or PBS (N = 2) 3-7 days after giving birth. This was designated as study time 0. 48 h post-administration, breast-fed pups were intragastrically challenged with *V. cholerae* (1 x $10^5$ CFUs). **(B)** Data represents recovered CFUs of *V. cholerae* in the intestinal homogenates from each pup in the ZAC-3 IgG treatment (n = 46) and naïve groups (n = 11) across two independent experiments. **(C)** Data represents recovered CFUs of *V. cholerae* in the intestinal homogenates from each pup in the ZAC-3 IgA treatment (n = 34) and naïve groups (n = 11) across two independent experiments. Statistical analysis was performed using unpaired t test without Welch's correction after confirming that variances were equal. Panel A was created in BioRender: Adewunmi, Y. (2026) https://BioRender.com/7tgdxtz.

with IgG concentrations progressively increasing over the course of the study. Furthermore, ZAC-3 IgG but not IgA was detected in the serum (but not intestinal homogenates) of suckling pups at all time points tested. Complement-dependent vibriocidal activity was detected in serum and breast milk of mice treated with ZAC-3 IgG mRNA-LNP.

During the initial neonatal challenge study in which pups were separated from dams after gavage, no reduction in bacterial burden was observed in suckling mice following intragastric challenge with *V. cholerae*, an outcome we attribute to

insufficient levels of ZAC-3 antibodies in the intestinal lumen. In contrast, when pups were returned to dams for continuous breastfeeding following challenge, a significant reduction in intestinal *V. cholerae* burden was observed in pups exposed to ZAC-3 IgG, whereas only a modest, non-significant reduction was observed in pups exposed to ZAC-3 IgA via breast milk. These findings indicate that sustained antibody delivery through ongoing milk intake can increase luminal antibody availability in the presence of a target, with IgG achieving levels sufficient to mediate statistically relevant protection in this model.

The absence of detectable ZAC-3 in intestinal homogenates under non-challenge conditions suggests that milk-delivered antibodies may not remain in the intestinal lumen in the absence of antigen. In suckling mice, antibodies present in milk can be rapidly absorbed into circulation, consistent with our detection of ZAC-3 IgG in pup serum. Together, these observations support a model in which luminal antibody availability is dynamic and influenced not only by delivery through milk but also by antigen engagement within the intestinal lumen (i.e., through the presence of a cognate target), which may promote functional antibody retention and protection.

Our results reinforce that the importance of breastfeeding for newborns cannot be overemphasized. These findings are consistent with a study in a rotavirus model, which showed that antibodies acquired by pups from breast milk, rather than via placental transfer, conferred protection through the continued presence of pathogen-specific antibodies in the intestinal lumen, where milk-delivered antibodies could act locally in the gut [23].

Importantly, given the presence of ZAC-3 IgG in the serum of pups, our data further highlight the potential utility of maternal mRNA-mediated antibody delivery for systemic neonatal infections when maternal immunization occurs after delivery, as milk-derived antibodies can enter the neonatal circulation and thus provide a route for postnatal passive immunization. Thus, our findings also extend impact beyond gastrointestinal infections.

During intestinal development in infants, there are two main mechanisms for the systemic absorption of antibodies and other macromolecules. One such method is gut permeability, where protein or drug permeability is size restricted (up to 70 kDa) and depends on the distance of the tissue from the stomach [24]. This form of transport typically occurs before gut closure, a process that occurs around 7 days in human infants and 3 weeks in neonatal mice. The other mechanism involves the active transport of IgG and other immune complexes from mucosal tissues into circulation through binding to FcRn [25]. In rodents, this mechanism is used exclusively to maintain adequate levels of IgG in the blood over extended periods of time [26]. Although FcRn-mediated transport is one possible mechanism, our study did not directly evaluate this pathway, and additional work would be required to determine the specific processes involved. Understanding these transport mechanisms will be important for optimizing mRNA-based maternal immunization platforms, particularly for pathogens requiring mucosal protection. Future studies dissecting these pathways will help guide rational improvements in antibody isotype selection, dosing, and delivery strategies.

Pregnancy and lactation are known to be associated with major physiological changes that can influence antibody distribution and pharmacokinetics, including antibody transfer to the placenta and breast milk [27,28]. In this study, we observed that lactating dams exhibited significantly lower serum ZAC-3 IgG and IgA concentrations compared to non-lactating mice receiving the same mRNA-LNP dose. In parallel, ZAC-3 IgG and IgA was detectable in the breast milk of lactating dams at concentrations representing at least one-fifth of those measured in serum.

Similar patterns have been reported in other settings; for example, IgG levels following SARS-CoV-2 vaccination are consistently higher in maternal serum than in paired breast milk samples from the same individual [29]. Despite these observations, the biodistribution dynamics of mRNA vaccines – and especially mRNA-mediated antibody therapies – during pregnancy and lactation remain poorly characterized [30].

Evidence from the literature further suggests that vaccine-derived mRNA can transiently cross the blood–breast milk barrier in humans, where it is present at low concentrations and is rapidly cleared, with no evidence of translational activity in breast milk [31]. If similar dynamics were present in our model, such differences in biodistribution could contribute to the altered serum antibody levels observed in lactating dams. However, this study was not designed to determine whether reduced serum antibody concentrations reflect redistribution, altered clearance, or accumulation of mRNA in specific tissues.

Notably, based on estimated maternal blood volume and milk antibody concentrations, the daily amount of ZAC-3 IgG and IgA detected in breast milk (approximately 0.5 – 4 µg/mL) does not fully account for the difference in total circulating antibody levels between lactating and non-lactating mice. This observation suggests that additional extravascular compartments or physiological factors associated with lactation may influence systemic antibody concentrations. Elucidating these mechanisms will require dedicated pharmacokinetic and tissue-distribution studies.

IgA provides protection at mucosal surfaces and can be delivered to sites such as the intestinal lumen and breast milk, where it contributes to bacterial neutralization through mechanisms including agglutination [32,33]. IgA mAbs also exhibit enhanced antigen binding and neutralization capacity against enteric and respiratory pathogens compared to IgG of the same clone [34], and prior work from our group has demonstrated that mRNA-encoded dIgA can be produced in situ and trafficked to mucosal tissues in other models [16]. The IgA2 mRNA construct used in this study encoded the IgA heavy chain (HC), light chain (LC), and joining chain (JC), consistent with designs intended to support dimeric IgA assembly. Our data showed that IgA2 expression was approximately ten-fold lower than IgG1 in both serum and breast milk and was not detectable in intestinal tissues of either dams or pups, despite targeted screening for mucosal localization. Because IgA2 levels were substantially lower, we did not further assess J-chain incorporation, polymerization state, or secretory IgA formation, as insufficient material was available to meaningfully characterize its form in this model.

These findings are consistent with established principles governing IgA delivery to milk. In rodents, IgA present in breast milk is derived predominantly from plasma cells localized to mammary tissue, with relatively inefficient transfer of circulating IgA into milk. Quantitative studies in other species support this distinction; for example, in pigs, approximately 40% of colostral IgA is serum-derived, whereas in mature milk nearly 90% of IgA originates from local mammary synthesis rather than systemic circulation [35]. Together, these observations suggest that systemically expressed IgA may be intrinsically limited in its ability to achieve functionally relevant concentrations in milk and intestinal tissues beyond the colostral phase. Achieving higher mucosal IgA levels via mRNA-based delivery may require further optimization at multiple levels. These may include approaches that increase the magnitude and duration of IgA expression, such as increasing mRNA dose or dosing frequency. In addition, influencing the distribution of the expressed antibody through choice of administration route to favor tissues with access to mucosal transport mechanisms, which could also enhance the access of systemically produced IgA to pathways that enable efficient delivery to luminal compartments.

The use of an mRNA-encoded antibodies has applications beyond enteric disease models, particularly for systemic infections in infants, including neonatal sepsis, pneumonia, and meningitis. Moreover, this study represents, to our knowledge, the first demonstration of maternal transfer of an mRNA-encoded antibody to pups. While our findings show the presence of antibodies in the days following maternal mRNA immunization, the underlying mechanisms driving this persistence remain unclear. Without direct data on mRNA stability or antibody half-life, it is difficult to distinguish whether the presence of these antibodies days after immunization is due to the continuous translation of the mRNA or to inherently long-lived IgG or IgA antibodies. Future studies should aim to delineate these mechanisms by tracking mRNA degradation kinetics and quantifying antibody half-life in maternal and neonatal compartments. Such insights would strengthen our understanding of the platform's potential for long-term protection and inform rational design of mRNA-based strategies for passive immunization. Notably, the expressed IgG and IgA retained both antigen binding and IgG in-vitro neutralization activity, supporting its functional relevance.

## Acknowledgments

We thank Dr. Graham Willsey for technical assistance and for valuable feedback and Grace Freeman-Gallant for statistical consultation. We are grateful to the Wadsworth Center's Veterinary Sciences staff for assistance in oversight of maternal-pup studies. We also thank the Wadsworth Center's Tissue Culture and Media core facility for bacterial culture media.

## Author contributions

**Conceptualization:** Jennifer E. Doering, Yetunde Adewunmi, Cailin E. Deal, Obadiah Plante, Andrea Carfi, Nicholas J Mantis.

**Formal analysis:** Jennifer E. Doering, Yetunde Adewunmi, Nicholas J Mantis.

**Funding acquisition:** Cailin E. Deal, Andrea Carfi, Nicholas J Mantis.

**Investigation:** Jennifer E. Doering, Yetunde Adewunmi, Cailin E. Deal.

**Methodology:** Jennifer E. Doering, Yetunde Adewunmi, Cailin E. Deal.

**Project administration:** Cailin E. Deal, Obadiah Plante, Andrea Carfi, Nicholas J Mantis.

**Resources:** Nicholas J Mantis.

**Supervision:** Obadiah Plante, Nicholas J Mantis.

**Writing – original draft:** Jennifer E. Doering, Yetunde Adewunmi, Cailin E. Deal, Nicholas J Mantis.

**Writing – review & editing:** Jennifer E. Doering, Yetunde Adewunmi, Cailin E. Deal, Nicholas J Mantis.

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
