## [Decision Letter · Decision Letter 0]

27 Oct 2025

Delivery of a protective Vibrio cholerae-specific, mRNA-derived monoclonal IgG into mouse breast milk

Dear Dr. Mantis,

Thank you for submitting your manuscript to PLOS Neglected Tropical Diseases. Your manuscript has now been reviewed by three experts in the field. Each reviewer found the work to be novel and significant. Nevertheless, each also asked for revisions. I hope that you find the reviews helpful in drafting a revised manuscript. Their comments are appended here for your review. After careful consideration, we feel that it has merit but does not fully meet PLOS Neglected Tropical Diseases's publication criteria as it currently stands. Therefore, we invite you to submit a revised version of the manuscript that addresses the points raised during the review process.

Please submit your revised manuscript within 60 days Dec 26 2025 11:59PM. If you will need more time than this to complete your revisions, please reply to this message or contact the journal office at plosntds@plos.org. Please include the following items when submitting your revised manuscript:

We look forward to receiving your revised manuscript.

Kind regards,

Jim

James Michael Fleckenstein, M.D.

Academic Editor

Elsio Wunder Jr

Section Editor

Shaden Kamhawi

co-Editor-in-Chief

Paul Brindley

co-Editor-in-Chief

**Additional Editor Comments (if provided):**

**Journal Requirements:**

**Reviewers' Comments:**

Reviewer's Responses to Questions

**Key Review Criteria Required for Acceptance?**

**Methods**

-Are the objectives of the study clearly articulated with a clear testable hypothesis stated?

-Is the study design appropriate to address the stated objectives?

-Is the population clearly described and appropriate for the hypothesis being tested?

-Is the sample size sufficient to ensure adequate power to address the hypothesis being tested?

-Were correct statistical analysis used to support conclusions?

-Are there concerns about ethical or regulatory requirements being met?

Reviewer #1: -The objectives are novel and well motivated, but the central hypothesis (that maternal mRNA-LNP administration can confer mucosal protection to pups) is not explicitly framed as testable.

-The design demonstrates maternal-to-infant antibody transfer but does not directly test protective efficacy against V. cholerae, leaving a gap between objectives and outcomes.

-The use of dams and pups is appropriate, but more detail on animal numbers, age, and selection criteria would strengthen reproducibility.

-Sample sizes are modest; statistical power is not discussed, making it unclear whether negative findings (e.g., lack of intestinal protection) reflect biology or underpowered experiments.

-Analyses appear standard, but details are limited (e.g., methods for multiple comparisons, justification of tests). Greater transparency is needed to assess robustness.

-Ethical approval is mentioned, but more detail on animal welfare monitoring and adverse effect assessment would provide reassurance.

Reviewer #2: see below

Reviewer #3: (No Response)

**Results**

-Does the analysis presented match the analysis plan?

-Are the results clearly and completely presented?

-Are the figures (Tables, Images) of sufficient quality for clarity?

Reviewer #1: -Broadly yes, but the study leaves key mechanistic questions unresolved (e.g., FcRn role, SIgA formation, antibody persistence). Additional experiments would better align results with the stated goals.

-The main findings are clearly shown, but presentation is incomplete. Absolute antibody concentrations are missing, protective efficacy is not directly demonstrated, and mechanistic claims sometimes exceed the data.

-Figures are generally clear, but should include absolute concentrations (µg/mL) and more precise labeling to enhance reproducibility and interpretation.

Reviewer #2: see below

Reviewer #3: (No Response)

**Conclusions**

-Are the conclusions supported by the data presented?

-Are the limitations of analysis clearly described?

-Do the authors discuss how these data can be helpful to advance our understanding of the topic under study?

-Is public health relevance addressed?

Reviewer #1: -Partially. The data convincingly demonstrate maternal-to-infant transfer of mRNA-encoded IgG; however, claims of protective efficacy are overstated, given the lack of intestinal protection or challenge studies.

Are the limitations of the analysis clearly described? Some are acknowledged (e.g., uncertainty about antibody persistence), but key gaps—such as the failure to generate SIgA, the absence of protective benchmarks, and limited mechanistic insight—require a fuller discussion.

Do the authors discuss how these data can contribute to advancing our understanding of the topic under study? Yes. They highlight the novelty of mRNA-LNP–driven antibody transfer via milk; however, the discussion could be improved by better connecting mechanistic insights (e.g., trafficking, FcRn role) to the translational potential.

Is public health relevance addressed? Yes, the potential for infant protection against emerging pathogens is emphasized, but claims should be tempered until efficacy is demonstrated in relevant challenge models.

Reviewer #2: see below

Reviewer #3: (No Response)

**Editorial and Data Presentation Modifications?**

Reviewer #1: (No Response)

Reviewer #2: see below

Reviewer #3: (No Response)

**Summary and General Comments**

Reviewer #1: The manuscript explores whether maternal delivery of mRNA-LNPs encoding mAbs can generate protective titters in breast milk and transfer immunity to offspring. Using the anti-V. cholerae antibody ZAC-3, the authors demonstrate the systemic and milk expression of mRNA-encoded IgG and the maternal-to-infant transfer of this antibody. To date, this is the first direct evidence of mRNA-encoded antibodies passing from mother to pups. The study is well-written, conceptually novel, and relevant, particularly for infant protection against emerging pathogens. However, important clarifications, additional data, and more nuanced interpretations are needed. While it establishes proof of concept, it does not demonstrate protective efficacy against enteric challenge, limiting translational claims. The lack of intestinal protection highlights the need for optimized antibody formats (e.g., SIgA) and a deeper understanding of antibody trafficking and persistence. The manuscript is suitable for publication after major revision, with emphasis on tempering claims of efficacy, clarifying IgA biology, and strengthening data on antibody localization and function.

Major concerns

1-The study shows detectable ZAC-3 IgG in maternal milk and pup serum, but not in intestinal homogenates. This observation is critical: protection against V. cholerae requires mucosal antibodies, preferably secretory IgA. The hypothesis that FcRn-mediated transport sequesters IgG into the systemic compartment at the expense of luminal protection is plausible, but requires direct evidence (e.g., antibody quantification in intestinal contents at different ages, FcRn knockout controls, or blocking FcRn function). Without such data, the mechanistic conclusion remains speculative.

2-The finding of much lower expression of IgA2 versus IgG1 highlights a major challenge. However, the study does not address whether dimeric/secretory IgA was successfully formed, or whether constructs included J-chain or supported polymerization. Given the known superiority of SIgA in mucosal defense, demonstrating the feasibility of mRNA-mediated SIgA production (rather than monomeric IgA) is essential for the translational relevance of this platform to enteric infections.

3-The authors note uncertainty about whether sustained antibody detection reflects long-lived IgG versus ongoing translation from residual mRNA. This is an important mechanistic point and warrants direct investigation. Suggested approaches include: (a) Tracking mRNA degradation by RT-qPCR in tissues. (b) Measuring antibody decay rates after single vs. repeated administrations.(c) Without such data, the persistence claims remain incomplete.

4-The study does not establish the quantitative thresholds of mucosal antibody required to block colonization. Passive transfer experiments with purified ZAC-3 IgG/IgA at defined doses (administered by oral gavage to pups) would help establish protective levels.

Minor concerns

- The manuscript should clearly distinguish between the detection of antibodies in serum/milk and protective efficacy in vivo. Currently, the narrative risks overinterpreting findings as proof of protection when it was not achieved.

- The discussion of neonatal gut permeability vs. FcRn transport is appropriate but should be framed as a hypothesis rather than a firm conclusion.

· It would be valuable to clarify whether any adverse effects were observed in dams or pups following mRNA-LNP administration (e.g., inflammation, growth delay).

- Figures and tables should include absolute concentrations (µg/mL), not just relative values, to facilitate reproducibility and inter-study comparison.

Reviewer #2: Very interesting

To my knowledge, first use of mRNA technology to try to induce antigen-specific Ab in breast milk to protect newborns (in this case pups)

Long known that vaccinating moms can boost breat milk Ig, both IgG and in primed moms/previously infected, IgA

In this study, the attempt was to induce high level responses in breast milk in not primed

used a cholera model

Pros: non invasive bacterial pathogen, breast milk protects, there was a well characterized mAb

Cons: cholera doesn’t teally affect under 18-24 months (in part becuase in cholera endemic areas, breast milk is the source of hydration, not contaminated water

Bu tfo rthe purpose of a model system, perfectly reasonable

The researchers achieved significant IgG levels in moms/dams, less so IgA

IgG was in breast milk

Made it into pups

The IgG had vibriocidal activity (suggesting correct confirmation)

in the model system, did not change the CFU of intestinal V cholerae in challenged pups

A few thoughts:

the manuscript would be strengthen by discussing/adressing/clarifying below

Why give IV, why not IM for more prolonged expression

IV not practical in emergency situation

What cell type is making

Most Ig in breast milk is produced locally in breast tissue, why would you think IV would give breast milk Ig

Any evidence your IgA actually made with JC, and any staining of breast tissue for production?

Define HC, LC, JC for reader

My memory is rodents pump dIgA from blood into biliary system; does same happen with breast?

why did you give a single VLP with both an IgG and an dIgA mRNA

Why not due separately?

p12 ELISA protocol, seems like decribed IgG capture/detection, please add the IgA methods

line 379 missing ‘antibody”

Why do WC Elisa and not LPS

Judging protection used presence of V cholerae in intestine, but no evidence anti-LPS antibodoes kill the bacteria so the assay may not be the optimal one

Fig 2demonstrates a profound effect of lactation status, would discuss more in discussion

the milking mice protocol was a tour de force

Figures only show IgG, please add IgA

Reviewer #3: This is a well-written and thoughtfully designed study that explores the feasibility of using mRNA-LNP technology to express and deliver pathogen-specific monoclonal antibodies (mAbs) through breast milk to newborns. The authors use a Vibrio cholerae model and the well-characterized ZAC-3 monoclonal antibody, providing clear experimental evidence of successful expression in maternal serum and milk, and transfer to pups. The manuscript addresses an important gap in maternal-infant immunology and proposes an innovative platform for passive immunization in outbreak or vaccine-limited settings. The approach is conceptually strong, and the experimental framework is well executed. However, some aspects of experimental design, data interpretation, and mechanistic depth could be strengthened to increase the impact and clarity of the work.

Major Weaknesses and Recommendations

1. Although functional antibodies were detected in maternal serum, milk, and pups, no reduction in V. cholerae colonization was observed. This discrepancy warrants further investigation. The authors could consider using a higher dose, repeated mRNA administrations, or an alternative route of administration to achieve protective antibody levels.

2. The authors hypothesize FcRn-mediated transfer but do not directly evaluate it. Including supporting experiments (e.g., FcRn blockade, knockout models, or tracking antibody localization) or providing a more detailed discussion of how FcRn-mediated transport might limit gut availability would strengthen the manuscript.

3. IgA expression was low and only briefly analyzed. Given the mucosal protective role of IgA, the authors should provide more quantitative details and consider discussing potential approaches to optimize IgA expression or delivery.

4. Some figures rely on pooled samples or have small n, which limits statistical power. The authors should provide exact n values for each experiment in the figure legends and display individual data points to enhance transparency and statistical robustness.

5. The study is innovative and relevant for outbreak response; however, the translational implications are underdeveloped. Expanding the discussion on potential clinical applications, safety considerations, and how this platform could integrate with existing maternal vaccination strategies would strengthen the manuscript’s impact.

Also, the current title “Delivery of a protective Vibrio cholerae-specific, mRNA-derived monoclonal IgG into mouse breast milk” is potentially misleading, as the study does not demonstrate actual protection against V. cholerae challenge in neonatal mice. For accuracy and transparency, it would be more appropriate to revise the title to reflect the demonstrated findings rather than the intended goal.

While antibody expression and transfer were clearly demonstrated, the lack of mucosal protection highlights critical challenges in achieving protective antibody titers at the mucosal surface. Addressing these mechanistic gaps, particularly regarding FcRn-mediated transport and mucosal localization, would significantly strengthen the manuscript and enhance its translational impact. The study is conceptually strong and technically solid but would benefit from additional analysis or expanded discussion to explain the lack of protective effect and to better define the translational significance of the findings.

PLOS authors have the option to publish the peer review history of their article (what does this mean? ). If published, this will include your full peer review and any attached files.

**Do you want your identity to be public for this peer review?** For information about this choice, including consent withdrawal, please see our Privacy Policy .

Reviewer #1: No

Reviewer #2: No

Reviewer #3: **Yes:** Alaullah Sheikh

**Figure resubmission:**
---

## [Editor Report · Decision Letter 1]

29 Jan 2026

Dear Dr. Mantis,

We are pleased to inform you that your manuscript 'Maternal expression and breast milk transfer of an mRNA- encoded monoclonal antibody in a murine model of cholera' has been provisionally accepted for publication in PLOS Neglected Tropical Diseases.

Best regards,

James Michael Fleckenstein, M.D.

Academic Editor

Elsio Wunder Jr

Section Editor

Shaden Kamhawi

co-Editor-in-Chief

Paul Brindley

co-Editor-in-Chief

---

## [Editor Report · Acceptance letter]

Dear Dr. Mantis,

We are delighted to inform you that your manuscript, "

Maternal expression and breast milk transfer of an mRNA- encoded monoclonal antibody in a murine model of cholera," has been formally accepted for publication in PLOS Neglected Tropical Diseases.

Best regards,

Shaden Kamhawi

co-Editor-in-Chief

Paul Brindley

co-Editor-in-Chief
